

# Integrable Floquet systems related
# to logarithmic conformal field theory

**Vsevolod I. Yashin[1⋆], Denis V. Kurlov[2,3], Aleksey K. Fedorov[2,3] and Vladimir Gritsev[4,2]**

**1** Steklov Mathematical Institute of Russian Academy of Sciences,
Gubkina str., 8, Moscow 119991, Russia
**2** Russian Quantum Center, Skolkovo, Moscow 143025, Russia
**3** National University of Science and Technology "MISIS",
Moscow 119049, Russia
**4** Institute for Theoretical Physics, Universiteit van Amsterdam,
Science Park 904, Postbus 94485, 1090 GL Amsterdam,
The Netherlands

⋆ viyashin@protonmail.com

## Abstract

We study an integrable Floquet quantum system related to lattice statistical systems in the universality class of dense polymers. These systems are described by a particular non-unitary representation of the Temperley-Lieb algebra. We find a simple Lie algebra structure for the elements of Temperley-Lieb algebra which are invariant under shift by two lattice sites, and show how the local Floquet conserved charges and the Floquet Hamiltonian are expressed in terms of this algebra. The system has a phase transition between local and non-local phases of the Floquet Hamiltonian. We provide a strong indication that in the scaling limit this non-equilibrium system is described by the logarithmic conformal field theory.



# 1 Introduction

Integrable systems play a tremendous role in our understanding of many-body statistical classical and quantum systems. A great number of conceptual insights had emerged from the notable examples of exactly solvable models. For instance, solution of the two-dimensional (2D) Ising model by Onsager [1] has eventually led to the concepts of scaling and universality. In addition, the 2D Ising model became a benchmark for the renormalization group technique and various numerical methods. Later on, a multi-state generalization of the Ising model, the so-called Potts model, has been solved in some cases [2] and has revealed a great amount of interesting mathematics, e.g., Tutte and chromatic polynomials from graph theory and the Temperley-Lieb algebras [3], just to mention a few. A particular case of the latter one is the central object of this paper.

The Temperley-Lieb (TL) algebra (formally defined below in Section 3) has one free parameter $\beta$ which eventually defines its representations. Many *different* realizations of the TL algebra in terms of physically-interesting objects can have the same value of $\beta$. In particular, for $\beta = \sqrt{2}$ there is a representation related to the quantum Ising chain, while for the representation that corresponds to the isotropic Heisenberg spin-1/2 chain (XXX model) one has $\beta = 2$. Here we are concerned with the case of $\beta = 0$. The TL generators in this case have a representation in terms of the supersymmetric spin chain related to the $\mathfrak{gl}(1|1)$ algebra [4]. It is a well-known fact [4,5] that the continuum limit of this spin chain provides a realization for the logarithmic conformal field theory (LOG-CFT) with the central charge $c = -2$. This field theory appears in the scaling limit of critical dense polymers [6,7]. This LOG-CFT is also a theory of the so-called symplectic fermions introduced in Refs. [8,9]. The structure of the TL algebra at $\beta = 0$, its continuum limit, and the algebraic structure of the continuum theory have been intensively studied in a series of works [10–14], see also [15] and [16] for a nice overview of these developments.

Motivated by the historical line of thoughts on the importance of integrable models, we

introduce an integrable quantum Floquet dynamics [17] (see also [18] for the recent developments) with the aim to understand periodically driven many-body systems exactly. The two-step protocol described below in Section 2 has a very close resemblance with integrable lattice models in the brick-wall-like representation of Baxter [19]. Indeed, after an appropriate analytic continuation, the logarithm of the transfer matrix can be identified with a *quantum Floquet Hamiltonian*, defined below (Section 2). Obviously, the transfer matrix of a classical lattice model is a non-local object. This preclude an immediate writing down of analytic expression for the Floquet Hamiltonian. Using the map outlined above the Floquet Hamiltonian can be expressed in terms of an infinite number of conserved charges.

The problem of finding all the conserved charges for a generic integrable Floquet protocol based on the TL algebra seems to be intractable.[1] In the recent paper [20] the conserved charges for anisotropic Heisenberg model have been computed in terms of the TL generators (in the basis of irreducible words on the algebra). To construct a logarithm of the transfer matrix one should sum up these charges with powers of a formal parameter, but this seems intractable at the moment. We note that our construction presented in this paper is different and is motivated by the Floquet construction and by application of Lie-algebraic techniques.

In the present work we demonstrate that for $\beta = 0$ the conserved charges and the Floquet Hamiltonian can be computed in the closed form. Furthermore, we show that conserved charges lie inside an infinite dimensional $\mathfrak{sl}(2)$ loop algebra, which could be useful to better understand the LOG-CFT at $c = -2$. For the representation in terms of symplectic fermions one can diagonalize the Floquet Hamiltonian exactly, see Section 4. In addition, we find some sort of a phase transition, which is related to the convergence of the series defining the Floquet Hamiltonian. We are tempted to interpret it in terms of the locality-nonlocality transition, similar to the case of the Floquet $XY$ model [21].

## 2 Integrable Floquet dynamics

We study systems with periodic alteration between two Hamiltonians $\mathcal{H}_e$ and $\mathcal{H}_o$ that act for duration $T_1$ and $T_2$, correspondingly. The total period of the system is $T = T_1 + T_2$. This (the so-called two-step) protocol is a quite generic setup describing a Floquet (time-periodic) driven many-body quantum system,

$$\mathcal{H}(t) = \begin{cases} \mathcal{H}_e, & nT \leq t \leq nT + T_1, \\ \mathcal{H}_o, & nT + T_1 \leq t \leq nT + T_1 + T_2, \end{cases} \quad n \in \mathbb{Z}. \tag{1}$$

The two-step protocol (1) can be pictorially represented as shown in Fig. 1 (see also Ref. [17]). The stroboscopic time evolution of the system (1) is then governed by the operator $U_F$, given by

$$U_F = \exp(-iT_1 \mathcal{H}_e) \exp(-iT_2 \mathcal{H}_o) \equiv \exp(-iT \mathcal{H}_F), \tag{2}$$

where $\mathcal{H}_F$ is the effective *time-independent* Floquet Hamiltonian. In order to compute the Floquet Hamiltonian, one can use the Baker-Campbell-Hausdorff (BCH) formula:

$$\log \left( e^X e^Y \right) = X + Y + \frac{1}{2}[X, Y] + \frac{1}{12}\Big([X, [X, Y]] + [Y, [Y, X]]\Big) + \dots \tag{3}$$

However, in most cases it is impossible to sum the BCH series in a closed operatorial form. In this paper we present one of the rarest examples when this task can be accomplished.

---

[1]However, several lowest charges can be obtained quite easily. We would like to thank Prof. Jesper Lykke Jacobsen for interesting communications on this point.

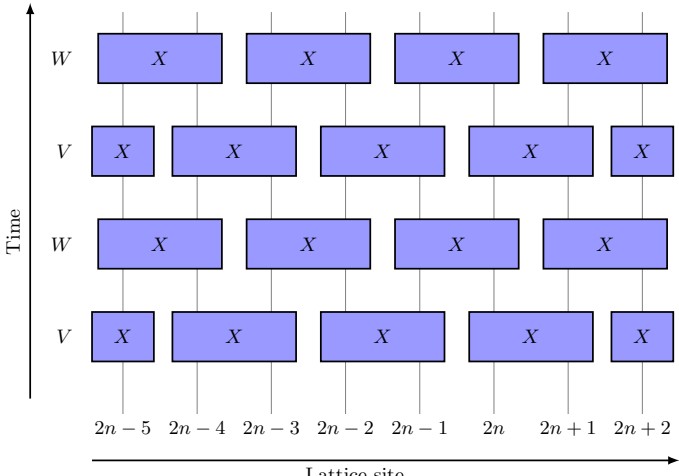

Figure 1: Brick-wall protocol. A space-time picture of the lattice version of the integrable two-step Floquet protocol for the Floquet dynamics. System evolves for time $T_1$ with a Hamiltonian $\mathcal{H}_e$ (for the layer $V$) and for the time $T_2$ with a Hamiltonian $\mathcal{H}_o$ (for the layer $W$). Layers $V$ and $W$ correspond to the evolution operators $\exp(-iT_1\mathcal{H}_e)$ and $\exp(-iT_2\mathcal{H}_o)$ respectively. Here $X$ denotes an operator acting between two neighboring lattice cites while satisfying the Yang-Baxter equation. A particular case of the latter is provided by the TL algebra, e.g. by the $\mathrm{TL}_N(0)$.

We remind that the operator $\mathcal{Q}$ is called a *conserved charge* if it commutes with the Hamiltonian $\mathcal{H}$, thus being a time-independent quantity. One of the definitions of quantum integrability (see Ref. [22] for extensive discussion on the notions of integrability in quantum systems) is that the Hamiltonian system is considered to be *integrable* if it wields the complete set of mutually commuting charges $\{\mathcal{Q}_n\}$. In this case it is possible to fully characterize the evolution of a system, namely to find its eigensystem. Similarly, the Floquet-system is called *Floquet-integrable*, if it contains the full set of (a sufficient number of) operators $\{Q_n\}$ that commute with the Floquet Hamiltonian $H_F$ (equivalently, with the Floquet evolution operator $U_F$). These operators have stroboscopic-time conservation: they form conserving family at times $mT$, where $m \in \mathbb{Z}$. Some general considerations about Floquet integrable models may be found in [17].

We should mention here that our two-step Temperley-Lieb algebraic Floquet protocol is conjectured to be integrable for some special points in the $T_1-T_2$ parameter space, in particular for $T_1 = T_2$ for generic $\beta$ [18]. However, in this paper we show that for $\beta = 0$ the two-step protocol is integrable regardless of what the values of $T_1$ and $T_2$ are.

## 3 Temperley-Lieb algebra and commuting Floquet charges

In this Section we first define our construction in terms of the TL algebra. Then we establish an infinite-dimensional loop algebra for the charges that commute with the Floquet Hamiltonian and each other.

### 3.1 Floquet protocol in terms of the Temperley-Lieb algebra

The Temperley-Lieb algebra $\mathrm{TL}_N(\beta)$ is an associative algebra that appears frequently in the context of various integrable models [23]. The algebra contains a free parameter $\beta \in \mathbb{C}$ and is

generated by the elements $\{e_i\}_{i=1}^{N-1}$ that satisfy the relations:

$$
\begin{aligned}
e_i^2 &= \beta e_i\,, \\
e_i e_{i\pm1} e_i &= e_i\,, \\
e_i e_j &= e_j e_i\,, \qquad |i-j| > 1\,.
\end{aligned}
\tag{4}
$$

We are interested in the case of $\beta = 0$ that is related to the dimer representations of the Temperley-Lieb algebra [23, 24]. In order to impose periodic boundary conditions and deal with translationally invariant systems, we include an additional generator $e_0 \equiv e_N$ that satisfies $e_0^2 = \beta e_0 = 0$ and the additional relations

$$
e_0 e_J e_0 = e_0\,, \quad e_J e_0 e_J = e_J\,, \qquad\qquad J \in \{1, N-1\}\,.
\tag{5}
$$

The resulting algebra generated by $\{e_i\}_{i=0}^{N-1}$ is called the *periodic Temperley-Lieb algebra* and denoted by $\mathrm{pTL}_N(0)$, see e.g. Refs. [25–27] for review. The systems we are interested in are often invariant under the shift by two sites, therefore we require that $N$ is even. Also note that we aim to consider the thermodynamic limit, $N \to \infty$.

In terms of $\mathrm{pTL}_N(0)$, the Hamiltonians $\mathcal{H}_\mathrm{e}$ and $\mathcal{H}_\mathrm{o}$ from Eq. (1) are written as

$$
\mathcal{H}_\mathrm{e} = \sum_{i=0}^{(N-2)/2} e_{2i} \equiv \sum_{i \text{ even}} e_i\,, \quad \mathcal{H}_\mathrm{o} = \sum_{i=0}^{(N-2)/2} e_{2j+1} \equiv \sum_{i \text{ odd}} e_i\,,
\tag{6}
$$

and the Floquet evolution operator is given by Eq. (2). In this identification the Floquet time evolution looks like a brick-wall protocol, see Fig. 1.

Note that the algebraic structure of $\mathrm{pTL}_N(0)$ (here, $\beta = 0$ is essential) is preserved under the following automorphisms

$$
e_i \mapsto \begin{cases} t e_i\,, & \text{for } i \text{ even}\,, \\ t^{-1} e_i\,, & \text{for } i \text{ odd}\,, \end{cases} \quad t \in \mathbb{C} \setminus \{0\}\,.
\tag{7}
$$

Let us denote

$$
\tau = \sqrt{T_1 T_2}\,, \qquad z = -i\tau\,.
\tag{8}
$$

The number $\tau$ can be understood as the "averaged" period of the protocol, and $z$ is its Wick rotation. Thus, for later convenience we redefine the generators $e_i$ using the automorphism (7) with the parameter $t = \sqrt{T_2/T_1}$. Therefore, we are interested in examining the Floquet evolution operator of form

$$
U_F(z) = \exp(z\mathcal{H}_\mathrm{e}) \exp(z\mathcal{H}_\mathrm{o}) = \exp(z\mathcal{H}_F(z))\,.
\tag{9}
$$

Let us also mention that the average Hamiltonian $\mathcal{H}$ equals $\mathcal{H}_F$ in the Trotter limit

$$
\mathcal{H} = \mathcal{H}_\mathrm{e} + \mathcal{H}_\mathrm{o} = \lim_{z \to 0} \mathcal{H}_F(z)\,.
\tag{10}
$$

## 3.2 Lie algebraic structure of Temperley-Lieb algebra at $\beta = 0$

The algebra $\mathrm{pTL}_N(0)$ has a number of nice properties. In particular, it turns out to have a rather convenient Lie algebra of commutators, see Appendix A for further details. We denote the Lie algebra of commutators as

$$
\mathfrak{tl} = \mathrm{Lie}\Big(\mathrm{pTL}_N(0)\Big)\,.
\tag{11}
$$

Let us introduce the generators $\mathfrak{q}_i^m$, which correspond to the Lie polynomials of degree $m$ labelled by the lattice site $i$:

$$
\mathfrak{q}_i^0 = 1\,, \quad \mathfrak{q}_i^1 = e_i\,, \quad \mathfrak{q}_i^m = [e_i, [e_{i+1}, \cdots, [e_{i+m-2}, e_{i+m-1}]\cdots]]\,.
\tag{12}
$$

The generators $\mathfrak{q}_i^m$ span the algebra $\mathfrak{tl}$. One should be careful with the fact that $\mathfrak{q}_i^m$ are defined only for $m < N$. However, in the case $N \to \infty$ that we are interested in, this does not lead to confusion.

Then, let us consider the subalgebra $\mathfrak{tl}_\pm \subset \mathfrak{tl}$ of generators invariant under the shift by two lattice sites. It consists of generators $\mathfrak{q}_+^m$ and $\mathfrak{q}_-^m$ defined as

$$\mathfrak{q}_+^m = \sum_i \mathfrak{q}_i^m, \quad \mathfrak{q}_-^m = \sum_i (-1)^i \mathfrak{q}_i^m. \tag{13}$$

For our purposes, it is also convenient to use the following basis:

$$\mathfrak{q}_e^m = \frac{1}{2}(\mathfrak{q}_+^m + \mathfrak{q}_-^m) = \sum_{i \text{ even}} \mathfrak{q}_i^m, \quad \mathfrak{q}_o^m = \frac{1}{2}(\mathfrak{q}_+^m - \mathfrak{q}_-^m) = \sum_{i \text{ odd}} \mathfrak{q}_i^m. \tag{14}$$

Indeed, in terms of the operators (14) the Hamiltonians $\mathcal{H}, \mathcal{H}_e, \mathcal{H}_o$ are given by

$$\mathcal{H} = \mathfrak{q}_+^1, \quad \mathcal{H}_e = \mathfrak{q}_e^1, \quad \mathcal{H}_o = \mathfrak{q}_o^1. \tag{15}$$

One can check (see Appendix A) that the following identities hold

$$\begin{aligned}
[\mathcal{H}_e, \mathcal{H}_o] &= \mathfrak{q}_-^2, \quad [\mathcal{H}_\alpha, \mathfrak{q}_+^{2s+1}] = 0, \quad [\mathcal{H}_\alpha, \mathfrak{q}_\alpha^{2s+1}] = 0, \\
[\mathcal{H}_e, \mathfrak{q}_o^{2s+1}] &= \mathfrak{q}_-^{2s+2} + \mathfrak{q}_-^{2s}, \qquad \quad [\mathcal{H}_o, \mathfrak{q}_e^{2s+1}] = -(\mathfrak{q}_-^{2s+2} + \mathfrak{q}_-^{2s}), \\
[\mathcal{H}_e, \mathfrak{q}_-^2] &= -2(\mathfrak{q}_e^3 + 2\mathfrak{q}_e^1), \qquad \quad [\mathcal{H}_o, \mathfrak{q}_-^2] = 2(\mathfrak{q}_o^3 + 2\mathfrak{q}_o^1), \\
[\mathcal{H}_e, \mathfrak{q}_-^{2s}] &= -2(\mathfrak{q}_e^{2s+1} + \mathfrak{q}_e^{2s-1}), \quad [\mathcal{H}_o, \mathfrak{q}_-^{2s}] = -2(\mathfrak{q}_o^{2s+1} + \mathfrak{q}_o^{2s-1}),
\end{aligned} \tag{16}$$

where $s > 0$ and $\alpha \in \{e, o\}$. Let us then introduce additional operators

$$\tilde{\mathfrak{q}}_\beta^m = \sum_{l=0}^{\lfloor \frac{m-1}{2} \rfloor} \binom{m-1}{m-1-l} \mathfrak{q}_\beta^{m-2l}, \qquad \beta \in \{e, o, +, -\}. \tag{17}$$

The operators (17) are very convenient for examining the structure of Lie algebra $\mathfrak{tl}_\pm$ (see Appendix A). Note that the subalgebra $\{\tilde{\mathfrak{q}}_+^{2s+2}\}_s$ is a center, i.e. these operators commute with all elements in $\mathfrak{tl}_\pm$. One can also check that the three subalgebras $\{\tilde{\mathfrak{q}}_e^{2s+1}\}_s, \{\tilde{\mathfrak{q}}_o^{2s+1}\}_s, \{\tilde{\mathfrak{q}}_-^{2s+2}\}_s$ are commutative and maximal. Now, let us define

$$H^c = \tilde{\mathfrak{q}}_-^{2c}, \qquad E^a = \tilde{\mathfrak{q}}_e^{2a+1}, \qquad F^b = \tilde{\mathfrak{q}}_o^{2b-1}, \tag{18}$$

where $a = 0, 1, \dots$ and $b, c = 1, 2, \dots$. Quite remarkably, the operators (18) turn out to satisfy the relations for the $\mathfrak{sl}(2)$ loop algebra:

$$\begin{aligned}
[H^n, H^m] &= 0, & [E^n, E^m] &= 0, & [F^n, F^m] &= 0, \\
[H^n, E^m] &= 2E^{n+m}, & [H^n, F^m] &= -2F^{n+m}, & [E^n, F^m] &= H^{n+m},
\end{aligned} \tag{19}$$

which is a central result of this subsection. Thus, we have obtained that the Lie algebra $\mathfrak{tl}_\pm$ is decomposed into a center $\{\tilde{\mathfrak{q}}_+^{2s+2}\}_s$ and an algebra $\text{Lie}(\mathcal{H}_e, \mathcal{H}_o)$, which is a subalgebra of the $\mathfrak{sl}(2)$ loop algebra (a subalgebra of elements with positive loop parameters), and one has

$$\mathcal{H}_e = E^0, \qquad \mathcal{H}_o = F^1. \tag{20}$$

Finally, the loop algebra relations (19) may be expressed in terms of $\{\mathfrak{q}_\pm^n\}_n$ so as to give

$$\begin{aligned}
[\tilde{\mathfrak{q}}_+^n, \tilde{\mathfrak{q}}_+^m] &= 0, & &\text{for any } n, m, \\
[\tilde{\mathfrak{q}}_+^n, \tilde{\mathfrak{q}}_-^m] &= 0, & &\text{for even } n \text{ and any } m, \\
[\tilde{\mathfrak{q}}_+^n, \tilde{\mathfrak{q}}_-^m] &= -2\tilde{\mathfrak{q}}_-^{n+m}, & &\text{for odd } n \text{ and any } m, \\
[\tilde{\mathfrak{q}}_-^n, \tilde{\mathfrak{q}}_-^m] &= 0, & &\text{for } n, m \text{ both even or both odd}, \\
[\tilde{\mathfrak{q}}_-^n, \tilde{\mathfrak{q}}_-^m] &= 2\tilde{\mathfrak{q}}_+^{n+m}, & &\text{for even } n \text{ and odd } m.
\end{aligned} \tag{21}$$

## 3.3 Floquet conserved charges

Using the commutation relations presented in Eqs. (16)–(19) here we find the set of local charges for the Hamiltonian $\mathcal{H}$ and the evolution operator $U_F$.

### 3.3.1 Charges of the average Hamiltonian

Eq. (21) directly implies that the set of commuting charges for the average Hamiltonian $\mathcal{H} = q_+^1$ in Eq. (10) is given by[2]

$$\mathcal{Q}_m = \mathfrak{q}_+^m. \tag{22}$$

The charges $\mathcal{Q}_m$ are referred to as *higher Hamiltonians* in Ref. [10].

**Remark 1.** *Note that if we disregard the boundary conditions, then this system has a boost operator $\mathcal{B} = \sum_j j e_j$, such that*

$$[\mathcal{Q}_m, \mathcal{B}] = m\mathcal{Q}_{m+1} + (m-2)\mathcal{Q}_{m-1}. \tag{23}$$

### 3.3.2 Charges of the evolution operator

**Proposition 1.** *There is a set of local charges[3] for the evolution operator $U_F(z)$, given by*

$$\begin{aligned}
Q_m &= \tilde{\mathfrak{q}}_+^m, && \text{if } m \text{ even}, \\
Q_m &= \tilde{\mathfrak{q}}_+^m + \frac{z}{2}\tilde{\mathfrak{q}}_-^{m+1}, && \text{if } m \text{ odd}.
\end{aligned} \tag{24}$$

*Proof.* First, it is trivial to show that $U_F$ commutes with $Q_m$ for even m. Indeed, we know [see Eq. (21)], that the even charges $Q_m = \tilde{\mathfrak{q}}_+^m$ commute with *any* element in the algebra, which obviously includes $U_F$. Now, suppose $m$ is odd. Clearly, the requirement $[U_F, Q_m] = 0$ is equivalent to

$$e^{-z\,\mathrm{ad}_{\mathcal{H}_e}} Q_m = e^{z\,\mathrm{ad}_{\mathcal{H}_o}} Q_m. \tag{25}$$

Using Eqs. (14), (18), and (24), one can easily see that in terms of the generators of the $\mathfrak{sl}(2)$ loop algebra the conserved charges read

$$Q_{2s+1} = E^s + F^{s+1} + \frac{z}{2}H^{s+1}. \tag{26}$$

Then, keeping in mind that $\mathcal{H}_e = E^0$, $\mathcal{H}_o = F^1$, and using the following relations:

$$\begin{aligned}
\mathrm{ad}_{E^0} Q_{2s+1} &= H^{s+1} - zE^{s+1}, & \mathrm{ad}_{E^0}^2 Q_{2s+1} &= -2E^{s+1}, & \mathrm{ad}_{E^0}^3 Q_{2s+1} &= 0, \\
\mathrm{ad}_{F^1} Q_{2s+1} &= -H^{s+1} + zF^{s+2}, & \mathrm{ad}_{F^1}^2 Q_{2s+1} &= -2F^{s+2}, & \mathrm{ad}_{F^1}^3 Q_{2s+1} &= 0,
\end{aligned} \tag{27}$$

from Eqs. (25) and (26) we immediately obtain

$$e^{-z\,\mathrm{ad}_{E^0}} Q_{2s+1} = e^{z\,\mathrm{ad}_{F^1}} Q_{2s+1} = E^s + F^{s+1} - \frac{z}{2}H^{s+1}, \tag{28}$$

so that Eq. (25) is satisfied. It is also a straightforward check that all the charges $Q_m$ commute with each other. $\qquad\square$

Note that in the Trotter limit $z \to 0$ the charges of the Floquet evolution operator and those of the average Hamiltonian are equivalent to each other, as expected.

---

[2]Here, we equivalently could have taken $\tilde{\mathfrak{q}}_+^m$ instead of $\mathfrak{q}_+^m$.

[3]We conjecture that this set is also complete.

**Remark 2.** *We note that the expression for the first commuting charge*

$$Q_1 = E^0 + F^1 + \frac{z}{2}H^1 = \sum_{j=0}^{N-1} e_j - \frac{i\tau}{2}\sum_{j=0}^{N-1}(-1)^j\left[e_j, e_{j+1}\right], \tag{29}$$

*as follows from Eq. (26), coincides* [4] *with the the* $\beta = 0$ *limit of the corresponding expression derived for generic* $\beta$ *and* $T_1 = T_2 = T$ *[so that in Eq. (29)* $\tau = T$ *] in a recent paper [18].*

### 3.4 Floquet Hamiltonian

Using the Baker-Campbell-Hausdorff (BCH) formula one can write the Floquet Hamiltonian $\mathcal{H}_F$ as the series expansion

$$\log U_F(z) = z\mathcal{H}_F(z) = \log\left(e^{z\mathcal{H}_e}e^{z\mathcal{H}_o}\right) = \sum_{k=1}^{\infty} z^k Z_k, \tag{30}$$

where $Z_k$ are the Lie polynomials of degree $k$ made of Hamiltonians $\mathcal{H}_e, \mathcal{H}_o$. Then, taking into account the structure of the Lie algebra $\mathfrak{tl}_\pm$, discussed in subsection 3.2, in particular its relation to the $\mathfrak{sl}(2)$ loop algebra, and using the symmetry

$$Z_k(X, Y) = (-1)^{k+1} Z_k(Y, X), \tag{31}$$

we find that the polynomial $Z_k$ has explicit form given in Proposition 2.

**Proposition 2.** *Lie polynomials* $Z_k$ *in the BCH expansion for the* $\mathfrak{sl}(2)$ *loop algebra are explicitly given by*

$$Z_{2s+1} = \frac{(-1)^s}{2s+1}\frac{1}{\binom{2s}{s}}\tilde{\mathfrak{q}}_+^{2s+1}, \qquad Z_{2s+2} = \frac{(-1)^s}{2s+2}\frac{1}{\binom{2s+1}{s}}\tilde{\mathfrak{q}}_-^{2s+2}, \quad s = 0, 1, \ldots \tag{32}$$

*Proof.* One can show that for the $\mathfrak{sl}(2)$ algebra with generators $\{H, E, F\}$ the following holds [28–30]:

$$\log\left(e^{zE}e^{zF}\right) = \frac{4\operatorname{arcsinh}\frac{z}{2}}{\sqrt{4+z^2}}\left(E + F + \frac{z}{2}H\right). \tag{33}$$

The series representation at $z = 0$ of this expression is

$$\log\left(e^{zE}e^{zF}\right) = \sum_{s=0}^{\infty}(-1)^s\left[\frac{z^{2s+1}}{2s+1}\frac{1}{\binom{2s}{s}}(E+F) + \frac{z^{2s+2}}{2s+2}\frac{1}{\binom{2s+1}{s}}H\right]. \tag{34}$$

This gives us all $Z_k$ in BCH series of the algebra $\mathfrak{sl}(2)$. Then, taking into account the integer loop label of the generators, the BCH expansion for the $\mathfrak{sl}(2)$ loop algebra takes the following form

$$\log\left(e^{zE^0}e^{zF^1}\right) = \sum_{s=0}^{\infty}(-1)^s\left[\frac{z^{2s+1}}{2s+1}\frac{1}{\binom{2s}{s}}(E^s+F^{s+1}) + \frac{z^{2s+2}}{2s+2}\frac{1}{\binom{2s+1}{s}}H^{s+1}\right]. \tag{35}$$

$\square$

---

[4]Note that the expression for $Q_1$ in Ref. [18] is derived for open boundary conditions, whereas here we are dealing with the periodic ones. While the expressions for the first conserved charge are the same in both cases (up to a trivial change of summation limits), this is no longer true for the higher order charges, since in the case of open boundary conditions there are also boundary terms present. Also note that because we count the lattice sites from zero, the term $\sim \sum_j(-1)^j[e_j, e_{j+1}]$ in Eq. (29) has a different sign from that in Ref. [18], where the sites are counted from one.

Therefore, we conclude that Floquet Hamiltonian has the following form

$$\log U_F(z) = z\mathcal{H}_F(z) = \sum_{s=0}^{\infty}(-1)^s\left[\frac{z^{2s+1}}{2s+1}\frac{1}{\binom{2s}{s}}\tilde{\mathfrak{q}}_+^{2s+1} + \frac{z^{2s+2}}{2s+2}\frac{1}{\binom{2s+1}{s}}\tilde{\mathfrak{q}}_-^{2s+2}\right]. \tag{36}$$

The first elements of the series (36) are given by

$$
\begin{aligned}
Z_1 &= \tilde{\mathfrak{q}}_+^1, & Z_2 &= \frac{1}{2}\tilde{\mathfrak{q}}_-^2, & Z_3 &= -\frac{1}{6}\tilde{\mathfrak{q}}_+^3, & Z_4 &= -\frac{1}{12}\tilde{\mathfrak{q}}_-^4, \\
Z_5 &= \frac{1}{30}\tilde{\mathfrak{q}}_+^5, & Z_6 &= \frac{1}{60}\tilde{\mathfrak{q}}_-^6, & Z_7 &= -\frac{1}{140}\tilde{\mathfrak{q}}_+^7, & \cdots,
\end{aligned}
\tag{37}
$$

We also note that the Floquet Hamiltonian can be expressed in terms of the odd charges of the Floquet evolution operator,

$$\log U_F(z) = \sum_{s=0}^{\infty}\frac{z^{2s+1}}{2s+1}\binom{2s}{s}^{-1}Q_{2s+1}. \tag{38}$$

**Remark 3.** *We note again that strictly speaking the charges $Q_m$ are defined only for $m < N$, therefore the above sums should be understood as exact expressions only when $N \to \infty$.*

**Remark 4.** *We note that the series may have only the finite radius of convergence $\mathcal{R}$. In the example considered in section 4, one has $\mathcal{R} = 1$. This is explained by the fact that the norm of the operator $Q_{2s+1}$ in equation (38) is approximately*

$$\|Q_{2s+1}\| \sim \binom{2s}{s}. \tag{39}$$

# 4 Charges and Floquet Hamiltonian in the representation of symplectic fermions

In this section we specify the relations found above to the model of symplectic fermions related to the $\mathfrak{gl}(1|1)$ spin chain. Note that similar analysis can also be applied to the dimer model representation [31] of the $\mathrm{TL}_N(0)$.

## 4.1 Symplectic fermions representation of the Temperley-Lieb algebra

We study the $\mathfrak{gl}(1|1)$ model with $T_1 = T_2$. The representation is defined as

$$e_i = (f_i^\times + f_{i+1}^\times)(f_i + f_{i+1}), \tag{40}$$

where the operator $f_i$ ($f_i^\times$) annihilates (creates) a so-called *symplectic fermion* on the $i$th lattice site. The operators $f_i, f_i^\times$ obey the following anticommutation relations

$$\{f_i, f_j\} = \{f_i^\times, f_j^\times\} = 0, \quad \{f_i, f_j^\times\} = (-1)^i\delta_{ij}. \tag{41}$$

We emphasise that $f_j$ and $f_j^\times$ are *not* Hermitian conjugates to each other. In terms of canonical fermionic creation and annihilation operators $c_j^\dagger$ and $c_j$, symplectic fermions are given by

$$f_j = i^j c_j, \quad f_j^\times = i^j c_j^\dagger, \tag{42}$$

where $i$ is the imaginary unit. One can show that any fermionic representation of the TL algebra with $\beta = 0$ that is bilinear in fermionic creation and annihilation operators and acts

nontrivially on two adjacent sites (i.e., $e_j$ acts only on sites $j$ and $j+1$) is equivalent to the representation (40) in terms of symplectic fermions, see Appendix B for the proof.

For later convenience, let us introduce the operators

$$b_j = f_j + f_{j+1}, \qquad b_j^\times = f_j^\times + f_{j+1}^\times, \tag{43}$$

which satisfy the following (non-canonical) anticommutation relations:

$$\{b_i, b_j\} = \{b_i^\times, b_j^\times\} = 0, \qquad \{b_i, b_j^\times\} = (-1)^i \left( \delta_{i,j+1} - \delta_{i,j-1} \right). \tag{44}$$

Then, using Eqs. (40) and (44) we immediately obtain that in the representation (40) the operators $\mathfrak{q}_i^m$ from Eq. (12) become

$$
\begin{aligned}
\mathfrak{q}_i^1 &= b_i^\times b_i, \\
\mathfrak{q}_i^{2s+1} &= (-1)^s (b_{i+2s}^\times b_i + b_i^\times b_{i+2s}), & (s > 0), \\
\mathfrak{q}_i^{2s+2} &= (-1)^{s+i} (b_{i+2s+1}^\times b_i - b_i^\times b_{i+2s+1}), & (s \geq 0).
\end{aligned}
\tag{45}
$$

Therefore, for the operators (13) one has

$$
\begin{aligned}
\mathfrak{q}_\pm^1 &= \sum_i (\pm 1)^i b_i^\times b_i, \\
\mathfrak{q}_\pm^{2s+1} &= (-1)^s \sum_i (\pm 1)^i (b_{i+2s}^\times b_i + b_i^\times b_{i+2s}), & (s > 0), \\
\mathfrak{q}_\pm^{2s+2} &= (-1)^s \sum_i (\mp 1)^i (b_{i+2s+1}^\times b_i - b_i^\times b_{i+2s+1}), & (s \geq 0).
\end{aligned}
\tag{46}
$$

Then, using the Fourier transform

$$b_j = \frac{1}{\sqrt{N}} \sum_{p \in BZ} e^{ipj} b_p, \qquad b_j^\times = \frac{1}{\sqrt{N}} \sum_{p \in BZ} e^{-ipj} b_p^\times, \tag{47}$$

where the sum is taken over Brillouin zone

$$BZ = \{0, \varepsilon, \dots, 2\pi - \varepsilon\}, \qquad \varepsilon = \frac{2\pi}{N}, \tag{48}$$

and the momenta are defined modulo $2\pi$, the charges $\mathfrak{q}_\pm^m$ in Eq. (46) can be written as

$$
\begin{aligned}
\mathfrak{q}_+^1 &= \sum_p b_p^\times b_p, & \mathfrak{q}_+^{2s+1} &= 2(-1)^s \sum_p \cos 2sp \, b_p^\times b_p, & (s > 0), \\
\mathfrak{q}_+^{2s+2} &= 2(-1)^{s+1} \sum_p \cos(2s+1)p \, b_{p-\pi}^\times b_p, & & (s \geq 0).
\end{aligned}
\tag{49}
$$

Likewise, for $\mathfrak{q}_-^m$ one obtains

$$
\begin{aligned}
\mathfrak{q}_-^1 &= \sum_p b_{p-\pi}^\times b_p, & \mathfrak{q}_-^{2s+1} &= 2(-1)^s \sum_p \cos 2sp \, b_{p-\pi}^\times b_p, & (s > 0), \\
\mathfrak{q}_-^{2s+2} &= 2i(-1)^{s+1} \sum_p \sin(2s+1)p \, b_p^\times b_p, & & (s \geq 0).
\end{aligned}
\tag{50}
$$

## 4.2 Floquet Hamiltonian in symplectic fermions representation

With the help of Eqs. (49) and (50), for the operators (17) we obtain

$$\tilde{\mathfrak{q}}_+^{2s+1} = \sum_p (2\sin p)^{2s} b_p^\times b_p, \qquad \tilde{\mathfrak{q}}_-^{2s+2} = -i \sum_p (2\sin p)^{2s+1} b_p^\times b_p, \quad (s \geq 0). \tag{51}$$

Therefore, using the general expression (36) of the Floquet Hamiltonian $\mathcal{H}_F$, in symplectic fermions representation we obtain

$$\mathcal{H}_F(z) = \sum_{s=0}^\infty (-1)^s \left[ \frac{z^{2s}}{2s+1} \frac{1}{\binom{2s}{s}} \tilde{\mathfrak{q}}_+^{2s+1} + \frac{z^{2s+1}}{2s+2} \frac{1}{\binom{2s+1}{s}} \tilde{\mathfrak{q}}_-^{2s+2} \right] = \sum_p \phi_p(z) b_p^\times b_p, \tag{52}$$

where the thermodynamic limit is assumed and we denoted

$$\phi_p(z) = \left( \frac{1 - iz\sin p}{1 + iz\sin p} \right)^{1/2} \frac{\operatorname{arcsinh}(z\sin p)}{z\sin p}. \tag{53}$$

Here $\phi_0(z) = \phi_\pi(z) = 1$, and Trotter limit corresponds to the average Hamiltonian $\mathcal{H}$:

$$\lim_{z\to 0} \phi_p(z) = 1, \qquad \lim_{z\to 0} \mathcal{H}_F(z) = \mathcal{H}. \tag{54}$$

Note that despite its form, the Hamiltonian (52) is *not* diagonal, since $[b_p^\times b_p, b_q^\times b_q] \neq 0$ for $p \neq q$. Moreover, $\mathcal{H}_F$ is not even diagonalisable since it is not normal, i.e. $[\mathcal{H}_F^\dagger, \mathcal{H}_F] \neq 0$. Nevertheless, one can bring it to the Jordan normal form.

## 4.3 Jordan normal form of the Floquet Hamiltonian

Let us now proceed with reducing the Floquet Hamiltonian (52) to its Jordan normal form. First of all, we rewrite the Floquet Hamiltonian (52) in terms of the Fourier components of symplectic fermions $f_p$ and $f_p^\times$. The latter are related to the operators $b_p$ and $b_p^\times$ as

$$b_p = (1 + e^{ip}) f_p, \qquad b_p^\times = (1 + e^{-ip}) f_p^\times, \tag{55}$$

where we used Eqs. (43) and (47). Thus, the Floquet Hamiltonian can be written as

$$\mathcal{H}_F(z) = \sum_{p=0}^{2\pi-\varepsilon} \frac{2}{z} \operatorname{arcsinh}(z\sin p) \cot\left(\frac{p}{2}\right) \left( \frac{1 - iz\sin p}{1 + iz\sin p} \right)^{1/2} f_p^\times f_p, \tag{56}$$

where we took into account that the summation over momenta is taken over the Brillouin zone (48).

Then, following Ref. [10] we introduce two fermionic modes [cf. Eq. (42)] which satisfy *canonical* anticommutation relations

$$\{c_{k,\sigma}, c_{q,\sigma'}\} = \{c_{k,\sigma}^\dagger, c_{q,\sigma'}^\dagger\} = 0, \qquad \{c_{k,\sigma}, c_{q,\sigma'}^\dagger\} = \delta_{k,q} \delta_{\sigma,\sigma'}, \tag{57}$$

where $\sigma, \sigma' \in \{+, -\}$ and $k \in \{0, \varepsilon, \dots, \pi - \varepsilon\}$, i.e. the Brillouin zone is "halved" as compared to Eq. (48). In the notations of Ref. [10] we have

$$c_{k,+} = \chi_k, \qquad c_{k,-} = \eta_k, \tag{58}$$

which we are going to use below.

Note that the square root in Eq. (56) requires extra care: the argument inside of the root may become negative in case $|\tau| > 1$. For this reason we examine the Floquet Hamiltonian in two separate regions.

### 4.3.1 The region $|\tau| \leq 1$

The Floquet Hamiltonian $\mathcal{H}_F$ in this case has a behaviour similar to a regular Hamiltonian $\mathcal{H}$, but with a deformed spectrum. In this region it holds that

$$\frac{1 + \tau \sin(p)}{1 - \tau \sin(p)} \geq 0, \tag{59}$$

therefore the Floquet Hamiltonian (56) can be written as

$$\mathcal{H}_F = \sum_{p=\varepsilon}^{\pi-\varepsilon} \frac{2}{\tau} \arcsin[\tau \sin(p)] \left\{ t_p(\tau) f_p^\times f_p + t_p^{-1}(\tau) f_{p-\pi}^\times f_{p-\pi} \right\} + 4 f_0^\times f_0, \tag{60}$$

where we separated zero modes, used $\tau = iz$ [see Eq. (8)], and introduced the coefficient

$$t_p(\tau) = \cot\left(\frac{p}{2}\right) \sqrt{\frac{1 + \tau \sin(p)}{1 - \tau \sin(p)}} \geq 0. \tag{61}$$

The fermionic modes $\chi_p$ and $\eta_p$ are related to the symplectic fermions in the following way

$$f_p = t_p^{-1/2}(\tau) \left( \frac{\chi_p + \eta_p}{\sqrt{2}} \right), \qquad\qquad f_{p-\pi} = t_p^{1/2}(\tau) \left( \frac{\chi_p - \eta_p}{\sqrt{2}} \right),$$

$$f_p^\times = t_p^{-1/2}(\tau) \left( \frac{\chi_p^\dagger - \eta_p^\dagger}{\sqrt{2}} \right), \qquad\qquad f_{p-\pi}^\times = t_p^{1/2}(\tau) \left( \frac{\chi_p^\dagger + \eta_p^\dagger}{\sqrt{2}} \right), \tag{62}$$

$$f_0 = \eta_0, \quad f_0^\times = \chi_0^\dagger, \quad f_\pi = \chi_0, \quad f_\pi^\times = \eta_0^\dagger.$$

The Floquet Hamiltonian (56) reduces to its Jordan normal form in terms of the canonical fermions. Explicitly, it reads

$$\mathcal{H}_F = \sum_{p=\varepsilon}^{\pi-\varepsilon} \frac{2}{\tau} \arcsin[\tau \sin(p)] \left( c_{p,+}^\dagger c_{p,+} - c_{p,-}^\dagger c_{p,-} \right) + 4 c_{0,+}^\dagger c_{0,-}, \tag{63}$$

where the operators $c_{p,\pm}$ are then related to $\chi_p$ and $\eta_p$ via Eq. (58). Note that in the limit $\tau \to 1$ the spectrum is piecewise linear (see Fig. 2).

### 4.3.2 The region $|\tau| > 1$

In this case, let us divide the momentum space into two intervals $\mathcal{I}_1 = \{p : |\tau \sin p| < 1\}$ and $\mathcal{I}_2 = \{p : |\tau \sin p| > 1\}$. Let us diagonalize the part of the Floquet Hamiltonian (56) with momenta inside of $\mathcal{I}_1$ by defining the two fermionic modes just as in Eq. (62). Note that the zero momentum mode always lies inside of $\mathcal{I}_1$.

For the part of Floquet Hamiltonian corresponding to the interval $\mathcal{I}_2$ some adjustment has to be made. In this interval we choose the branch $z = i\tau e^{-i\varepsilon}, \varepsilon \to 0$ corresponding to the inverse Wick rotation. Then, we obtain

$$\arcsin[\tau \sin(p)] = \frac{\pi}{2} - i \, \text{arccosh}[\tau \sin(p)],$$

$$\sqrt{\frac{1 + \tau \sin(p)}{1 - \tau \sin(p)}} = -i \left| \frac{1 + \tau \sin(p)}{1 - \tau \sin(p)} \right|^{1/2}. \tag{64}$$

Therefore, the part of the Floquet Hamiltonian (52) is given by

$$\sum_{p \in \mathcal{I}_2} \frac{2}{\tau} \arcsin[\tau \sin(p)] \left\{ -i t_p(\tau) f_p^\times f_p + i t_p^{-1}(\tau) f_{p-\pi}^\times f_{p-\pi} \right\}, \tag{65}$$

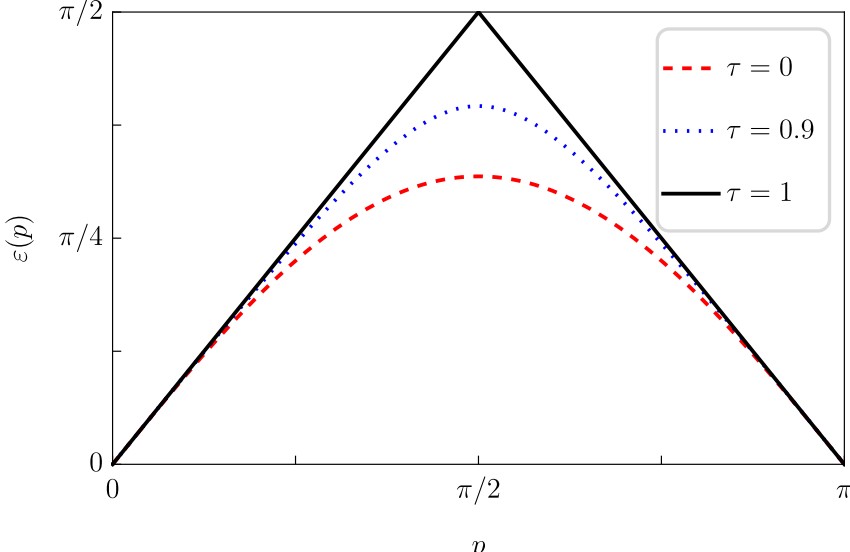

Figure 2: The Floquet Hamiltonian spectrum $\varepsilon(p) = \frac{1}{\tau} \arcsin[\tau \sin(p)]$ in the region $0 < p < \pi$, $0 < \tau < 1$. In the Trotter limit $\tau \to 0$ one has $\varepsilon(p) = \sin(p)$, while in the limit $\tau \to 1$ the spectrum is piecewise linear.

where this time the coefficient $t_p(\tau)$ reads

$$t_p(\tau) = \cot\left(\frac{p}{2}\right) \left| \frac{1 + \tau \sin(p)}{1 - \tau \sin(p)} \right|^{1/2} \geq 0, \tag{66}$$

and in the region $\mathcal{I}_2$ we define the $\chi_p, \eta_p$ fermions in the following way:

$$f_p = t_p^{-1/2}(\tau)\left(\frac{\chi_p + \eta_p}{\sqrt{2}}\right), \qquad f_{p-\pi} = t_p^{1/2}(\tau)\left(-i\frac{\chi_p - \eta_p}{\sqrt{2}}\right),$$

$$f_p^\times = t_p^{-1/2}(\tau)\left(i\frac{\chi_p^\dagger - \eta_p^\dagger}{\sqrt{2}}\right), \qquad f_{p-\pi}^\times = t_p^{1/2}(\tau)\left(\frac{\chi_p^\dagger + \eta_p^\dagger}{\sqrt{2}}\right). \tag{67}$$

We therefore once again obtain a Hamiltonian of the form (63). However, note that inside the interval $\mathcal{I}_2$ the spectrum becomes complex. We believe that this can be interpreted as a phase transition associated with the fact that in this regime the series (38) fails to converge. The question of analytic continuation is not considered here.

## 5  Discussion and outlook

We studied a particular realization of an integrable Floquet protocol corresponding to the case of periodic Temperley-Lieb algebra $\mathrm{pTL}_N(0)$. We found an underlying loop algebra structure of conserved charges for the evolution operator and obtained closed form expression for the Floquet Hamiltonian in terms of symplectic fermions.

The results of our analysis could perhaps be also interpreted in terms of nontrivial Floquet-integrable logarithmic conformal field theory. Indeed, the average Hamiltonian of our protocol is the LOG-CFT Hamiltonian discussed in the literature. Moreover the loop algebra structure of conserved charges is consistent with the LOG-CFT. It was recently observed that a large class of Floquet-driven CFTs are integrable in some sense [32–36]. It is therefore interesting to study further generalizations of these results to Log-CFT case.

We would like to emphasize that all previous studies of logarithmic CFTs were related to *equilibrium* statistical problems. On the contrary, we propose for the first time a realization of Log-CFT in the context of a *non-equilibrium*, Floquet driven system. The spectrum is linear at the points $q = 0, \pi$, which, combined with the affine algebra, clearly indicates that the system is a relativistic CFT. In addition, we propose an infinite family of conserved charges, which to the best of our knowledge is a new information in the context of Log-CFT (whether it is equilibrium or non-equilibrium).

Our results also point towards some sort of compact-noncompact phase transition in terms spreading of a support for the Floquet Hamiltonian, which is related to the divergence of a series expansion for the Floquet Hamiltonian.

It would be important also to generalize our approach to the case of the Temperley-Lieb algebras with arbitrary loop parameter $\beta$, corresponding e.g to the spin-1/2 $XXZ$-model. Even though we were not yet able to obtain the relations necessary for this type of analysis, some numerical experiments as well as alternative analytic approaches [20, 37] show some promise in this direction.

# Acknowledgements

The authors are grateful to D.S. Ageev, J.L. Jacobsen, Y. Miao, I.D. Motorin, and B. Nienhuis for useful discussions. The work of V.I.Y. was performed at the Steklov International Mathematical Center and supported by the Ministry of Science and Higher Education of the Russian Federation (agreement no. 075-15-2022-265). The work by V.G. is part of the DeltaITP consortium, a program of the Netherlands Organization for Scientific Research (NWO) funded by the Dutch Ministry of Education, Culture and Science (OCW). This study is also supported by the Russian Science Foundation (Grant No. 20-42-05002, work of D.V.K. and A.K.F.) and by the Priority 2030 program at the National University of Science and Technology MISIS.

# A   Lie algebraic structure of the $\mathrm{pTL}_N(0)$

In this Appendix we investigate the properties of Lie algebra generated by the commutators of $\mathrm{pTL}_N(0)$.

## A.1   Lie algebra of $\mathfrak{tl}$

**Property 1.** *By definition, $[e_i, e_{i+1}] = \mathfrak{q}_i^2$ and $[e_i, e_j] = 0$ if $|i - j| > 1$.*

**Property 2.** *The second-order commutators act as*

$$
\begin{aligned}
[[e_i, e_{i+1}], e_{i+1}] = [e_{i+1}, [e_{i+1}, e_i]] = -2e_{i+1}, \\
[[e_{i+1}, e_i], e_i] = [e_i, [e_i, e_{i+1}]] = -2e_i,
\end{aligned}
\tag{68}
$$

*therefore*

$$
\mathrm{ad}_{e_i}^3 e_j = 0.
\tag{69}
$$

*Proof.* Trivial computation. For example,

$$
[[e_i, e_{i+1}], e_{i+1}] = e_i e_{i+1} e_{i+1} - e_{i+1} e_i e_{i+1} - e_{i+1} e_i e_{i+1} + e_{i+1} e_{i+1} e_i = -2e_{i+1}.
\tag{70}
$$

$\square$

**Property 3.** *For any $h \in \mathrm{pTL}_N(0)$ the following holds*

$$\mathrm{ad}_{e_i}^2 h = -2e_i h e_i, \qquad \mathrm{ad}_{e_i}^3 h = 0. \tag{71}$$

*Proof.*

$$\mathrm{ad}_{e_i}^2 h = e_i(e_i h - h e_i) - (e_i h - h e_i)e_i = 0 - e_i h e_i - e_i h e_i + 0 = -2e_i h e_i, \tag{72}$$

$$\mathrm{ad}_{e_i}^3 h = -2e_i^2 h e_i + 2e_i h e_i^2 = 0. \tag{73}$$

$\square$

**Property 4.** *There is some freedom in the ways how to set up the brackets in $\mathfrak{q}_i^m$:*

$$[[\cdots[[e_i, e_{i+1}], e_{i+2}], \cdots], e_{i+m-1}] = [e_i, [e_{i+1}, \cdots, [e_{i+m-2}, e_{i+m-1}]\cdots]]. \tag{74}$$

*Proof.* Let us prove by induction on $m$. The cases $m = 1$ and $m = 2$ are trivial. The induction step is

$$
\begin{aligned}
&[e_i, [e_{i+1}, \cdots, [e_{i+m-2}, e_{i+m-1}]\cdots]] \\
&= [e_i, [[\cdots[[e_{i+1}, e_{i+2}], \cdots], e_{i+m-1}]] \\
&= /\text{Jacobi rule and } [e_i, e_{i+s}] = 0 \text{ if } s > 1/ \\
&= [[\cdots[[e_i, e_{i+1}], e_{i+2}], \cdots], e_{i+m-1}].
\end{aligned}
\tag{75}
$$

$\square$

**Property 5.** *It is possible to calculate the commutators $[e_j, \mathfrak{q}_i^m]$.*

$$
\begin{aligned}
[e_j, \mathfrak{q}_i^1] &= \delta_{j,i-1}\mathfrak{q}_{i-1}^2 - \delta_{j,i+1}\mathfrak{q}_i^2, \\
[e_j, \mathfrak{q}_i^2] &= \delta_{j,i-1}\mathfrak{q}_{i-1}^3 - 2\delta_{j,i}\mathfrak{q}_i^1 + 2\delta_{j,i+1}\mathfrak{q}_{i+1}^1 - \delta_{j,i+2}\mathfrak{q}_i^3, \\
[e_j, \mathfrak{q}_i^3] &= \delta_{j,i-1}\mathfrak{q}_{i-1}^4 + \delta_{j,i+1}(\mathfrak{q}_{i+1}^2 - \mathfrak{q}_i^2) - \delta_{j,i+3}\mathfrak{q}_i^4, \\
[e_j, \mathfrak{q}_i^m] &= \delta_{j,i-1}\mathfrak{q}_{i-1}^{m+1} + \delta_{j,i+1}\mathfrak{q}_{i+1}^{m-1} - \delta_{j,i+m-2}\mathfrak{q}_i^{m-1} - \delta_{j,i+m}\mathfrak{q}_i^{m+1}, \quad m > 3.
\end{aligned}
\tag{76}
$$

## A.2 Lie algebra of $\mathfrak{tl}_\pm$

**Property 6.** *By summation we conclude that*

$$
\begin{aligned}
[e_j, \mathfrak{q}_+^1] &= \mathfrak{q}_j^2 - \mathfrak{q}_{j-1}^2, \\
[e_j, \mathfrak{q}_+^2] &= \mathfrak{q}_j^3 - \mathfrak{q}_{j-2}^3, \\
[e_j, \mathfrak{q}_+^m] &= \mathfrak{q}_j^{m+1} - \mathfrak{q}_{j-m}^{m+1} + \mathfrak{q}_j^{m-1} - \mathfrak{q}_{j-m+2}^{m-1}, \qquad m \geq 3,
\end{aligned}
$$

$$
\begin{aligned}
[e_j, \mathfrak{q}_-^1] &= (-1)^j \left(-\mathfrak{q}_j^2 + \mathfrak{q}_{j-1}^2\right), \\
[e_j, \mathfrak{q}_-^2] &= (-1)^j \left(-\mathfrak{q}_j^3 - \mathfrak{q}_{j-2}^3 - 4\mathfrak{q}_j^1\right), \\
[e_j, \mathfrak{q}_-^m] &= (-1)^j \left(-\mathfrak{q}_j^{m+1} - (-1)^m \mathfrak{q}_{j-m}^{m+1} - \mathfrak{q}_j^{m-1} - (-1)^m \mathfrak{q}_{j-m+2}^{m-1}\right), \qquad m \geq 3.
\end{aligned}
$$

$$\tag{77}$$

Using this property, we obtain the relations (16).

**Property 7.** *All elements $\{\mathfrak{q}_\pm^{2s+1}, \mathfrak{q}_-^{2s+2}\}_s$ are generated as commutators of $\{\mathcal{H}_e, \mathcal{H}_o\}$, because*

$$
\begin{aligned}
[\mathcal{H}_e, \tilde{\mathfrak{q}}_e^{2s}] &= -2\tilde{\mathfrak{q}}_e^{2s+1}, & [\mathcal{H}_o, \tilde{\mathfrak{q}}_e^{2s}] &= 2\tilde{\mathfrak{q}}_o^{2s+1}, \\
[\mathcal{H}_e, \tilde{\mathfrak{q}}_e^{2s+1}] &= 0, & [\mathcal{H}_o, \tilde{\mathfrak{q}}_e^{2s+1}] &= -\tilde{\mathfrak{q}}_-^{2s+2}, \\
[\mathcal{H}_e, \tilde{\mathfrak{q}}_o^{2s+1}] &= \tilde{\mathfrak{q}}_-^{2s+2}, & [\mathcal{H}_o, \tilde{\mathfrak{q}}_o^{2s+1}] &= 0.
\end{aligned}
\tag{78}
$$

*Proof.* Directly follows from (16).  □

**Property 8.** *If $m$ is even, then $\mathfrak{q}_+^m$ commutes with $\mathcal{H}_e$ and $\mathcal{H}_o$, therefore also with all elements generated by them.*

**Property 9.** *All the elements $\mathfrak{q}_+^m$ commute.*

*Proof.* It was proven earlier in [10, 38].  □

Now, let us adopt the notation (18) and prove the relations (19).

**Property 10.** *The loop algebra relations (19) hold.*

*Proof.* Let us prove by induction in terms of the parameter $l = \min(n, m)$ for the commutators of (19). The base case $l = 0$ and $l = 1$ is essentially proven in Property 7. Now, let us suppose that relations (19) hold up to $\min(n, m) = l$. The induction step is easily proven via Jacobi relations. Let us give some examples: Suppose $n \le m$, then

$$
\begin{aligned}
[H^{n+1}, H^m] &= \left[[E^n, F^1], H^m\right] = \left[[E^n, H^m], F^1\right] + \left[E^n, [F^1, H^m]\right] \\
&= -2[E^{n+m}, F^1] + 2[E^n, F^{m+1}] = -2H^{n+m+1} + 2H^{n+m+1} = 0, \\
[E^{n+1}, E^m] &= \frac{1}{2}\left[[H^n, E^1], E^m\right] = \frac{1}{2}\left[[H^n, E^m], E^1\right] = [E^{n+m}, E^1] = 0, \\
[H^{n+1}, E^m] &= \left[[E^n, F^1], E^m\right] = \left[[E^m, F^1], E^n\right] = [H^{m+1}, E^n] = 2E^{n+m+1}, \\
[E^{n+1}, F^m] &= \frac{1}{2}\left[[H^n, E^1], F^m\right] = \frac{1}{2}\left[[H^n, F^m], E^1\right] + \frac{1}{2}\left[H^n, [E^1, F^m]\right] \\
&= -[F^{n+m}, E^1] + \frac{1}{2}[H^n, H^{m+1}] = H^{n+m+1}.
\end{aligned}
\tag{79}
$$

One can similarly verify the relations for all other cases.  □

## B  Symplectic fermions are unique

Let us consider some hypothetical representation $\xi(e)$ of a $\mathrm{TL}_N(0)$ (with open boundary conditions) that satisfies the following set of conditions:

1. $\xi(e)$ is quadratic in terms of fermionic operators,

2. $\xi(e)$ is local, i.e. $\xi(e_i)$ acts only on sites $i$ and $i+1$,

3. $\xi(e)$ is invariant under the shift by 2 sites.

We aim to prove that all representations satisfying the properties listed above lead to symplectic fermions. We believe that the third condition is generally not essential, but we use is for simplicity.

Let us denote

$$
\mathbf{c}_i = \left(c_i, c_i^\dagger, c_{i+1}, c_{i+1}^\dagger\right)^T.
\tag{80}
$$

A generic form of quadratic representations is given by

$$
\begin{aligned}
\xi(e_i) &= \frac{1}{2}\mathbf{c}_i^T G_e \mathbf{c}_i + v_e^T \mathbf{c}_i + \chi_e, \quad \text{if } i \text{ even}, \\
\xi(e_i) &= \frac{1}{2}\mathbf{c}_i^T G_o \mathbf{c}_i + v_o^T \mathbf{c}_i + \chi_o, \quad \text{if } i \text{ odd},
\end{aligned}
\tag{81}
$$

where $G_\alpha$ is some antisymmetric matrix corresponding to quadratic terms, $v_\alpha$ is a vector for linear terms, $\chi_\alpha$ is a constant.

The commutativity relation $e_i e_j = e_j e_i$, $|i - j| > 1$ leads to $v_e = v_o = 0$. The relation $e_i^2 = 0$ leads to $\chi_e = \chi_o = 0$. The remaining set of the Temperley-Lieb relations leads to the following *two* possible forms of $G_e$ and $G_o$:

## B.1 First case

$$
G_e = \alpha_e \begin{bmatrix} 0 & -1 & 0 & -t_e^{-1} \\ 1 & 0 & -t_e & 0 \\ 0 & t_e & 0 & 1 \\ t_e^{-1} & 0 & -1 & 0 \end{bmatrix}, \qquad
G_o = \alpha_o \begin{bmatrix} 0 & -1 & 0 & -t_o^{-1} \\ 1 & 0 & -t_o & 0 \\ 0 & t_o & 0 & 1 \\ t_o^{-1} & 0 & -1 & 0 \end{bmatrix},
\tag{82}
$$

$$
\alpha_e \alpha_o = -1 \,.
$$

The resulting generators of the TL factorize as

$$
\begin{aligned}
\xi(e_i) &= \alpha_e (c_i^\dagger + t_e^{-1} c_{i+1}^\dagger)(c_i - t_e c_{i+1}), && \text{if } i \text{ even}, \\
\xi(e_i) &= \alpha_o (c_i^\dagger + t_o^{-1} c_{i+1}^\dagger)(c_i - t_o c_{i+1}), && \text{if } i \text{ odd}, \\
\alpha_e \alpha_o &= -1 \,.
\end{aligned}
\tag{83}
$$

Let us introduce the symplectic fermions as

$$
\begin{aligned}
f_i &= x_i c_i, && f_i^\dagger = (-1)^i \frac{1}{x_i} c_i^\dagger, \\
\frac{x_{i+1}}{x_i} &= \begin{cases} -t_e, & \text{if } i \text{ even}, \\ -t_o, & \text{if } i \text{ odd}. \end{cases}
\end{aligned}
\tag{84}
$$

Here $x_i$ are some constants (depending on single variable $x_0$). Then the representation is expressed as

$$
\begin{aligned}
\xi(e_i) &= \alpha_e (f_i^\dagger + f_{i+1}^\dagger)(f_i + f_{i+1}), && \text{if } i \text{ even}, \\
\xi(e_i) &= \frac{1}{\alpha_e} (f_i^\dagger + f_{i+1}^\dagger)(f_i + f_{i+1}), && \text{if } i \text{ odd}.
\end{aligned}
\tag{85}
$$

The freedom of choosing the constants $\alpha_e$ and $1/\alpha_e$ can be eliminated by using the symmetry (7).

## B.2 Second case

$$
G_e = \alpha_e \begin{bmatrix} 0 & -1 & -t_e^{-1} & 0 \\ 1 & 0 & 0 & -t_e \\ t_e^{-1} & 0 & 0 & -1 \\ 0 & t_e & 1 & 0 \end{bmatrix}, \qquad
G_o = \alpha_o \begin{bmatrix} 0 & 1 & -t_o & 0 \\ -1 & 0 & 0 & -t_o^{-1} \\ t_o & 0 & 0 & 1 \\ 0 & t_o^{-1} & -1 & 0 \end{bmatrix},
\tag{86}
$$

$$
\alpha_e \alpha_o = -1 \,.
$$

The resulting generators of the TL factorize as

$$
\begin{aligned}
\xi(e_i) &= \alpha_e (c_i^\dagger + t_e^{-1} c_{i+1})(c_i - t_e c_{i+1}^\dagger), && \text{if } i \text{ even}, \\
\xi(e_i) &= \alpha_o (c_i^\dagger + t_o^{-1} c_{i+1})(c_i - t_o c_{i+1}^\dagger), && \text{if } i \text{ odd}, \\
\alpha_e \alpha_o &= -1 \,.
\end{aligned}
\tag{87}
$$

Let us introduce the symplectic fermions as

$$
\begin{aligned}
f_i &= x_i \begin{cases} c_i, & i \text{ even}, \\ c_i^\dagger, & i \text{ odd}, \end{cases} \\
f_i^\dagger &= (-1)^i \frac{1}{x_i} \begin{cases} c_i^\dagger, & i \text{ even}, \\ c_i, & i \text{ odd}, \end{cases} \\
\frac{x_{i+1}}{x_i} &= \begin{cases} -t_e, & \text{if } i \text{ even}, \\ -t_o, & \text{if } i \text{ odd}. \end{cases}
\end{aligned}
\tag{88}
$$

Here $x_i$ are some constants which depend on a single variable $x_o$. The representation is given by

$$\xi(e_i) = \alpha_e (f_i^\dagger + f_{i+1}^\dagger)(f_i + f_{i+1}), \quad \text{if } i \text{ even},$$
$$\xi(e_i) = \frac{1}{\alpha_e}(f_i^\dagger + f_{i+1}^\dagger)(f_i + f_{i+1}), \quad \text{if } i \text{ odd},$$

(89)

and the coefficients $\alpha_e$, $1/\alpha_e$ can be eliminated using the automorphism (7).

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
