# Peer review of "Integrable Floquet systems related to logarithmic conformal field theory"

_SciPost Physics, doi:SciPost Phys. 14, 084 (2023)_

## Round 1 · Referee Report · Anonymous (Referee 1) · 2022-8-17

Strengths

1- The authors provide an interesting new way of constructing conserved charges of the Floquet Hamiltonian of the periodic Temperley-Lieb algebra at $\beta = 0$. 2- The paper is well organised.

Weaknesses

1- Some claims are unproved. 2- There are a number of typos in the formulas. 3- There are a number of typos in the English.

Report

This paper investigates the integrable Floquet Hamiltonian related to the periodic Temperley-Lieb algebra in the special case where the loop weight is equal to zero. The authors nicely use Lie algebra techniques to construct an infinite number of charges that commute with this Hamiltonian. They also investigate the representation of this algebra related to the $g\ell(1,1)$ spin chain, or equivalently to the XX Hamiltonian. They express the conserved charges in terms of the symplectic fermions and argue that a phase transition arises in this model.

I find that the results are new and interesting and should eventually deserve a publication in Scipost. However, having gone over the calculations carefully, I find that there are a number of imprecisions and typos in the math which should be fixed before this paper is accepted. This paper also has a number of elementary typos in the English (for instance therms $\to$ terms, reson $\to$ reason, reions $\to$ regions, and many more), so I would ask that the authors do a proper spell-check before resubmitting their manuscript.

Here is my list of comments:

  • Equation (9) and the geometry presented in Figure 1 remind me very much of the construction used in arXiv:0812.2746 [math-ph] and in ref [31], with transfer matrices $T(u)$ built out of two successive rows of $R$-matrices shifted by one site. In fact, the Floquet evoluation operator $U_F(z) $ is precisely equal to this transfer matrix (with the correct parameterisation $z=z(u)$). In the above articles, it is shown that $T(u)$ is part of a two-parameter family of commuting transfer matrices, say $T(u,v)$, which commute only when the second parameter is varied. Thus performing an expansion of $T(u,v)$ along $v$, one obtains elements of the algebra that commute with $T(u)$. (These papers consider transfer matrices with a boundary, but the arguments also work for the periodic case.) It would be interesting to see if these conserved charges coincide with those of Section 3.3 for $\beta = 0$. Moreover, the construction of $T(u,v)$ and of the conserved charges also works away from $\beta = 0$, which could give some insight for the problem raised by the authors in the conclusion.

  • There seems to be a mistake with the very last equation in (16), at the bottom right. I find $[H_e, q_-^2] = -2 q_e^3 - 4 q_e^1$ $[H_o, q_-^2] = 2 q_e^3 + 4 q_e^1 $ and something slightly different for $q_-^{2n}$ with $n>1$.

  • The meaning of $tl_\pm$ is not clear. I had understood from the text above (13) that these are two separate algebras: $tl_+$ and $tl_-$, respectively generated by $q^m_+$ and $q^m_-$. But the text above (21) now appears to imply that $tl_\pm$ is a unique algebra.

  • A proof of equations (18) appears non trivial and is missing.

  • Clearly $[H_n,H_m]$ is missing as the left-hand side of the first equation in (20).

  • The sentence above (21) says that the Lie algebra is embedded into the $s\ell(2)$ loop algebra. But isn’t it the other way around?

  • The sentence starting 3.3 states that a complete set of conserved charges is constructed later in this section. This is an unsubstantiated claim: the authors indeed constructs many such charges in this section, but do not address the question of whether a full set of charges is indeed obtained.

  • Should the right side of (22) read $\tilde q^m_+$ instead of $q^m_+$?

  • I believe there is en error in the bottom equation of (49), where $i \sin$ should instead be replaced by $\cos$.

  • For (51), it would be useful to repeat the range of $s$, as I believe it starts at $s=1$ for the first equation and $s=0$ for the second.

  • The second member of (52) should be identical to (36), but it is not. On one hand, the powers of $z$ differ by one, and on another hand the first term has $q_-$ instead of $q_+$.

  • The reasoning behind the division into sections $R_1$ and $R_2$ and the resulting phase transition in section 4.3.2 is not sufficiently clear. My understanding is that these two regions describe different intervals of $p$ over which the sum is performed. So for instance (64) is incorrect: when $H_F$ contains terms of type $R_2$, it also has terms of type $R_1$. Then the transition is in fact one between $|z|<1$, which has only terms of type $R_1$, and $|z|>1$, which has terms of types $R_1$ and $R_2$. The authors should discuss these questions in greater detail.

  • In the abstract, the authors claim that they provide strong indication that their Floquet system is described by a logarithmic CFT. While the systems that they study are known to be related to dense polymers and symplectic fermions, both of which are related to $c=-2$ logarithmic CFTs, I don’t see any new elements provided by the authors’ calculation that give further information about the status of this model as a logarithmic CFT. The paragraph on the same topic in the conclusion is vague and lacks a proper argument that would convince me of this claim.

  • The authors should also explain why they believe that the phase transition has something to do with compactness vs non-compactness. As presented, I see no evidence pointing to this.

  • In Appendix A, a proof of Property 3 is missing. This proof seems non-trivial to me.

  • I believe the last equation in (75) to be incorrect: the second and last terms in the parenthesis should have the prefactor $(-1)^{m+1}$.

  • I don’t understand the relevance of Appendix B to this paper.

  • validity: good
  • significance: high
  • originality: good
  • clarity: good
  • formatting: good
  • grammar: below threshold

---

## Round 1 · Referee Report · Anonymous (Referee 2) · 2022-9-7

Strengths

New interesting results concerning particular integrable Floquet system

Weaknesses

Misprints, some calculations should be made more clear for general reader, unclear references to LogCFTs

Report

Report on ``Integrable Floquet systems related to logarithmic conformal field theory''

The paper ``Integrable Floquet systems related to logarithmic conformal field theory'' is devoted to the study of integrable Floquet quantum system described by a non-unitary representation of the Temperley-Lieb algebra. Authors construct an infinite number of charges that commute with this Hamiltonian and explicit operator expression for the Floquet Hamiltonian. The results are interesting and worth to be published in SciPost. However, I have some comments to be addressed, before the final decision could be made. Namely:

  1. When mentioning Floquet CFT it is neccesary to give the reference on the first paper on this topic (arXiv:1805.00031); Also, it is generally unclear what new information this study provides for understanding LogCFT (although the authors repeatedly mention their connection in the text).
  2. There are number of small misprints like in equation (20) (missing part on the left) and (22) (seems to be miss tilde). Also there are grammar typos in the text, so the spellcheck is required.
  3. I think it would be helpful to add some details on derivation of (16) and (18) (and probably other particular calculations which could be unclear for reader)

Requested changes

  1. When mentioning Floquet CFT it is neccesary to give the reference on the first paper on this topic (arXiv:1805.00031); Also, it is generally unclear what new information this study provides for understanding LogCFT (although the authors repeatedly mention their connection in the text).
  2. There are number of small misprints like in equation (20) (missing part on the left) and (22) (seems to be miss tilde). Also there are grammar typos in the text, so the spellcheck is required.
  3. I think it would be helpful to add some details on derivation of (16) and (18) (and probably other particular calculations which could be unclear for reader)

---

## Round 2 · Referee Report · Anonymous (Referee 2) · 2022-12-2

Report

I thank the authors for having addressed all the points raised in the report. I recommend this manuscript for publication on SciPost.

---

## Round 2 · Referee Report · Anonymous (Referee 1) · 2022-12-24

Report

The authors have made many changes following my previous report. I am overall happy with the changes, however there are still some minor remaining issues:

  • Clearly there is still a problem with the first equation of (16), and likewise with the first equation of (19).

  • Proposition 1 still refers to “the complete set of charges”. I already complained in my first report that this has not been proven. The authors should be more honest in the text, so that it is clear that the completeness is something that they believe, but that has not been established.

---

## Round 2 · Author Response

Firstly we would like to thank the Referees for the great assessment of our work. Below we
would like to answer the specific questions raised in the Referee report. For readability we
will first copy the original comment from the Referee (Remark), followed by our response
(Our Answer).

---

## Round 2 · List of Changes

To Referee A:

Remark: Equation (9) and the geometry presented in Figure 1 remind me very much of
the construction used in arXiv:0812.2746 [math-ph] and in ref [31], with transfer matrices
T (u) built out of two successive rows of R-matrices shifted by one site. In fact, the Floquet
evolution operator U_F(z) is precisely equal to this transfer matrix (with the correct
parametrization z = z(u)). In the above articles, it is shown that T(u) is part of a two-
parameter family of commuting transfer matrices, say T(u,v), which commute only when the
second parameter is varied. Thus performing an expansion of T(u,v) along v, one obtains
elements of the algebra that commute with T(u). (These papers consider transfer matrices
with a boundary, but the arguments also work for the periodic case.) It would be interesting
to see if these conserved charges coincide with those of Section 3.3 for β = 0. Moreover,
the construction of T(u,v) and of the conserved charges also works away from β = 0, which
could give some insight for the problem raised by the authors in the conclusion.
Our Answer: We thank the referee for the useful information and for the suggestion to
use a two-parametric transfer matrix for constructing commuting charges. We hope that it
might indeed be useful for further research, especially for going beyond the case of β = 0.
Obviously this requires separate investigation that we hope will be a subject of future works.
Remark: There seems to be a mistake with the very last equation in (16), at the bottom
right. I find [He, q2−] = −2q^3_e − 4q^1_e , [Ho, q^2_−] = 2q^3_e + 4q^1_e and something
slightly different for q^2_− with n > 1.

Answer: We are very grateful to the referee for very careful reading of our manuscript. In
the revised version all mistakes and misprints are corrected (we cross-checked the equations).

Remark: The meaning of tl_± is not clear. I had understood from the text above (13)
that these are two separate algebras: tl_+ and tl_-, respectively generated by
q^m_+ and q^m_-. But the text above (21) now appears to imply that tl_± is a
unique algebra.

Answer: The meaning of the statement may have been unclear due to a misprint (subalgebras → subalgebra).
The notation tl_± is used to indicate that it consists of the generators q_+ and q_−.

Remark: A proof of equations (18) appears non trivial and is missing.

Answer: We have improved the clarity of these calculations and moved these equations
to the Appendix A.

Remark: Clearly [H_n, H_m] is missing as the left-hand side of the first equation in (20).

Answer: Once again we thank the referee for the careful reading. We have added the
missing equations.

Remark: The sentence above (21) says that the Lie algebra is embedded into the sl(2)
loop algebra. But isn’t it the other way around?

Answer: We are making this statement more clear in the revised version: The Lie algebra
tl_± is decomposed into a center (commuting generators) made of even q_+ generators and
a ”positive part” of sl(2) loop algebra (involving only the generators with positive mode
numbers). In that sense it is a ”half” of the loop sl(2) algebra.

Remark: The sentence starting 3.3 states that a complete set of conserved charges is
constructed later in this section. This is an unsubstantiated claim: the authors indeed
constructs many such charges in this section, but do not address the question of whether a
full set of charges is indeed obtained.

Answer: The reviewer is correct that we do not have explicit proof of this claim, yet we
believe that it is true.

Remark: Should the right side of (22) read ̃q^m_+ instead of q^m_+ ?

Answer: We have added a remark after this equation.

Remark: I believe there is en error in the bottom equation of (49), where isin should
instead be replaced by cos.

Answer: We would like to thank our referee once more for careful reading. We have
corrected and updated these equations in the revised version.

Remark: For (51), it would be useful to repeat the range of s, as I believe it starts at
s = 1 for the first equation and s = 0 for the second.

Answer: Updated.

Remark: The second member of (52) should be identical to (36), but it is not. On one
hand, the powers of z differ by one, and on another hand the first term has q_− instead of
q^+.

Answer: There is a factor of z in front of the UF operator, so two equations have the
same powers of z. This is made explicit in the revised version.

Remark: The reasoning behind the division into sections R_1 and R_2 and the resulting
phase transition in section 4.3.2 is not sufficiently clear. My understanding is that these two
regions describe different intervals of p over which the sum is performed. So for instance
(64) is incorrect: when H_F contains terms of type R_2, it also has terms of type R_1. Then
the transition is in fact one between |z| < 1, which has only terms of type R1, and |z| > 1,
which has terms of types R1 and R2. The authors should discuss these questions in greater
detail.

Answer: We have rewritten this part and hope that now it is more clear.

Remark: In the abstract, the authors claim that they provide strong indication that their
Floquet system is described by a logarithmic CFT. While the systems that they study are
known to be related to dense polymers and symplectic fermions, both of which are related to
c=-2 logarithmic CFTs, I don’t see any new elements provided by the authors’ calculation
that give further information about the status of this model as a logarithmic CFT. The
paragraph on the same topic in the conclusion is vague and lacks a proper argument that
would convince me of this claim.
The authors should also explain why they believe that the phase transition has something
to do with compactness vs non-compactness. As presented, I see no evidence pointing to
this.

Answer: First of all we note that all previous studies of logarithmic CFTs were somehow
related to equilibrium statistical problems. In this paper, for the first time we propose a
realization of log-CFT in the context of a non-equilibrium, Floquet driven system. The
spectrum is linear at the points q = 0, π, which, combined with the affine algebra, clearly
indicates that the system is a relativistic CFT. Second, we propose an infinite family of
conserved charges, which to our knowledge is a new information in the context of log-CFT
(whether it is equilibrium or non-equilibrium). Third, the transition we found is related to
the divergence of a series expansion for the Floquet Hamiltonian.

Remark: In Appendix A, a proof of Property 3 is missing. This proof seems non-trivial
to me.

Answer: The proof is added in the current version.

Remark: I don’t understand the relevance of Appendix B to this paper.

Answer: We wanted to give a demonstration of how symplectic fermions appear from
a very general perspective. We hope this appendix could be retained for the reader to
familiarize more with this subject.
Remark: I believe the last equation in (75) to be incorrect: the second and last terms
in the parenthesis should have the prefactor (−1)^{m+1}.

Answer: We thank the reviewer for careful check of all the computations. All those
comments appear to be correct, and we regret that those errors appeared in the process of
writing.

To Referee B:

Remark: 1. When mentioning Floquet CFT it is neccesary to give the reference on
the first paper on this topic (arXiv:1805.00031); Also, it is generally unclear what new
information this study provides for understanding LogCFT (although the authors repeatedly
mention their connection in the text).

Our Answer: This work is cited as Ref. [32] in our text.

Remark: 2. There are number of small misprints like in equation (20) (missing part on
the left) and (22) (seems to be miss tilde). Also there are grammar typos in the text, so the
spellcheck is required.

Answer: We did our best to find and correct all typos and errors.

Remark: 3. I think it would be helpful to add some details on derivation of (16) and
(18) (and probably other particular calculations which could be unclear for reader)

Our Answer: We have improved the clarity of these calculations.

---

## Editorial Decision

published